# Familial Mediterranean Fever and Diet: A Narrative Review of the Scientific Literature

**DOI:** 10.3390/nu14153216

**Published:** 2022-08-05

**Authors:** Pasquale Mansueto, Aurelio Seidita, Marta Chiavetta, Dario Genovese, Alessandra Giuliano, Walter Priano, Antonio Carroccio, Alessandra Casuccio, Emanuele Amodio

**Affiliations:** 1Unit of Internal Medicine, Department of Health Promotion Sciences, Maternal and Infant Care, Internal Medicine and Medical Specialties (PROMISE), University of Palermo, 90127 Palermo, Italy; 2Unit of Internal Medicine, “V. Cervello” Hospital, Ospedali Riuniti “Villa Sofia-Cervello”, 90146 Palermo, Italy; 3Hygiene and Preventive Medicine Section, Department of Health Promotion Sciences, Maternal and Infant Care, Internal Medicine and Medical Specialties (PROMISE), University of Palermo, 90127 Palermo, Italy

**Keywords:** FMF, familial Mediterranean fever, diet, nutrition, review, autoinflammatory disease

## Abstract

Background: Familial Mediterranean fever (FMF) is an inherited autoinflammatory disease characterized by short acute attacks, with an as yet unknown cause. Several authors have investigated the role of some foods as potential triggers. This narrative review aims to analyze the correlation between diet and FMF clinical outcomes. Methods: The review was carried out following PRISMA statement guidelines, including all cross-sectional, case-crossover, and trial studies written in English and conducted between 1974 and 2022. Results: Overall, 642 records were identified through PubMed/MEDLINE (292) and Scopus (350), and seven studies were included: three out of seven (43%) studies evaluated FMF attack recurrence or time between consumption of high-fat foods and FMF attacks, while another three (43%) articles variously assessed FMF severity, and one (14%) evaluated the distribution of MEFV mutations. Conclusions: To date, conflicting results have been reported about fatty and salty food intake and FMF attack recurrence. Moreover, some authors have suggested a possible role of wheat. Finally, a diet rich in antioxidants and supplements with an anti-inflammatory effect could partially reduce symptoms and improve the well-being of FMF patients. Nevertheless, no conclusive data could be drawn about the impact of diet in FMF symptom triggering, and further studies are required to clarify this putative association.

## 1. Introduction

Known as the most common hereditary autoinflammatory disease, familial Mediterranean fever (FMF) is an autosomal recessive disorder, primarily occurring in populations from the eastern Mediterranean region, namely Turks, Jews, Arabs, and Armenians [1,2]. However, hundreds of patients in Europe, North America, and Japan have been reported, and this prevalence is expected to increase due to the massive population movements from endemic areas to these latter regions over the last two decades [3,4]. FMF prevalence varies from 1:500 to 1:1000 in endemic countries, with the highest reported prevalence of 1:395, in the Central Anatolia region [5].

Specifically, it is caused by a gain-of-function mutation in the MEFV gene, located in chromosome 16, which encodes a protein called pyrin. Pyrin is commonly expressed by neutrophils, as well as other cells belonging to the innate immune system: eosinophils, monocytes, and dendritic cells. The pyrin mutation might contribute to the activation of the pyrin inflammasome. Pyrin usually interacts with an adaptor protein, the apoptosis-associated speck-like protein (ASC), and this complex operates as a caspase-1 recruiter, creating the inflammasome, which activates pro-interleukin (IL)-1beta and pro-IL-18 in IL-1beta and IL-18. Furthermore, the pyrin inflammasome can activate the caspase-1 mediated cleavage of gasdermin D (GSDMD) pyroptosis (a peculiar type of cellular death characterized by cell swelling and lysis), with a further release of IL-1beta and IL-18. The overexpression and release of IL-1beta and IL-18 work as an initiator and amplifier of the innate immune response, which flows in an uncontrolled proinflammatory cytokine cascade, with consequential activation of NF-kB pathway and auto-amplification of inflammatory processes [2,6,7,8,9,10,11]. However, the single-gene model does not account for all FMF cases, and it is believed that other genetic and environmental factors could be involved in the etiopathogenesis.

Patients experience unpredictable, short and irregular episodes of fever and pain, with the typical characteristic of inflammation of the serous membranes, which can lead to hospitalization and, sometimes, even unnecessary surgery. These attacks usually resolve spontaneously within 24–72 h [12].

To achieve an acceptable quality of life and manage the disease, the current mainstay for treatment of FMF is colchicine. It has been demonstrated that its role in suppressing excessive monocyte activation is useful in reducing the severity and duration of symptoms as well as in preventing acute attacks and the development of complications, such as amyloidosis [13,14,15,16,17,18]. Several studies have reported that clinical bouts of FMF can be triggered by infection, trauma, psychological stress, excessive physical activity, exposure to cold, and menstruation [19,20].

Additionally, some foods have been suggested as potential triggers of FMF attacks (e.g., fatty and salty foods, cow’s milk, formulas, wheat) [21,22,23,24,25] and, for this reason, researchers are looking for a link between certain diet regimens and the severity of the clinical manifestations of FMF or patient response to colchicine therapy [21,22,23,24,25,26,27].

In the light of the above considerations, the aim of this narrative review was to provide, through a search of the literature, a synthesis of the results and findings of studies that investigated the effects of dietary behaviors on symptoms and treatment outcomes in FMF patients.

## 2. Materials and Methods

For this narrative review, the research group evaluated associations between FMF and many co-variates related to diet. The Preferred Reporting Items for Systematic Reviews and Meta-Analyses (PRISMA) guidelines were used to report the process and the results [28].

The main literature sources were PubMed/MEDLINE and Scopus, and the literature research was conducted on 2nd February 2022 by combining free text words and medical subject headings (MeSH). Each keyword was then combined using the Boolean operators “AND” and “OR”, obtaining the following search string:


*[(“familial Mediterranean fever” OR “MEFV” OR “FMF” OR “Mediterranean fever” OR “periodic fever syndrome”) AND (diet* OR food OR nutrition OR regimen OR habit OR wheat OR gluten OR milk OR nutrient OR egg OR fat)].*


### Inclusion/Exclusion Criteria

The inclusion criteria for the selected studies were: publication date between 1971 and 2022, English language, and analysis of the association between FMF and diet.

Only cross-sectional and case-crossover studies in English were included, regardless of the study country. All other study designs (such as book chapters, letters to the editor, commentaries, trial studies, prospective cohort studies, case-control studies, before and after studies, systematic review, narrative review, and meta-analysis) were excluded.

Articles were first screened (Screening phase) by two pairs of researchers. They independently read titles and abstracts and included an article on the basis of the inclusion criteria, thus allowing a double-check of the selected articles. In cases of discrepancy, two other, independent supervisory researchers reviewed the titles and abstracts, resolving each specific doubt.

Full texts of articles were downloaded only for the eligible studies and used for the Assessment phase. In this phase, each pair of researchers independently extracted data from the articles and compiled a Microsoft Excel spreadsheet. Any disagreement was resolved through confrontation within the same group research pair and, if the discrepancy persisted, the above-mentioned supervisory researchers compared the extracted data and resolved any divergences.

The collected variables were: year of publication, year(s) of study, geographic area, ethnicity, study design, primary objective of the study, number of participants, outcome(s) evaluated, type of food considered, type of association between FMF and type of food considered, findings of the study, and interpretations.

## 3. Results

### 3.1. Literature Search

As reported in Figure 1, in the first phase a total of 642 records were identified through the PubMed/MEDLINE (292) and Scopus (350) database searching platforms; among them there were 154 duplicates, which were removed. The final sample accounted for 488 unique papers.

From these, after preliminary screening, 478 records did not meet the pre-established inclusion criteria and, were thus excluded (details reported in Figure 1).

At this point 10 full-text articles were assessed for eligibility, and three of them were excluded for the following criteria: two were case-control studies (67%); the remaining one was not available for consultation (33%).

At the end of the assessment process, seven studies were included in the narrative review [21,22,23,24,25,26,27].

### 3.2. Characteristics of the Included Studies

The results are available in Table 1. Overall, six out of seven studies (86%) were published after 2009 [23,24,25,26,27,28], the remaining one in 1974 [21]. Similarly, in terms of the “year(s) of study” item, the results are superposable: four out of seven (57%) records evaluated data collected after 2015 [24,25,26,27], two out of seven (29%) articles collected data between 2007 and 2009 [22,23], and one out of seven (14%) collected data from 1960 to 1972 [23].

The following findings were obtained as regards ethnicity, study design and numerosity: Ethnicity: three out of seven (43%) collected data from Turkish patients [22,24,27], two out of seven (29%) worked with Armenian patients [21,23], one out of seven (14%) with Egyptians [26], and one out of seven (14%) analyzed Italian patients [25]; Study design: five out of seven records (71%) adopted a retrospective and/or a cross-sectional study design [21,22,24,26,27], while two out of seven articles (29%) selected a case-crossover study design [23,25]; Numerosity of the analyzed sample: only one article (14%) collected data from more than 1000 patients [27], while most of the studies (six out of seven, 86%) had smaller samples ranging from six to 167 patients [21,22,23,24,25,26].

As for the type of food considered, three out of seven (43%) articles focused on the adoption of a high-fat diet regimen [21,23,24], two out of seven (29%) evaluated the use of nutritional supplements [26,27], while the remaining one (14%) compared the assumption of mother’s milk, cow’s milk and formula milk [22], and another one (14%) studied the consumption of wheat [25].

In terms of outcomes, three (43%) studies evaluated FMF attack recurrence or the time between the consumption of high-fat foods and FMF attacks [21,23,26], another three (43%) articles variously assessed FMF severity [22,25,27], while one (14%) evaluated the distribution of MEFV mutations [24].

## 4. Discussion

Nowadays, the scientific literature strongly suggests that some diets, especially high-calorie ones, rich in simple sugars and saturated fatty acids, could promote a chronic proinflammatory state, and the concept of consuming antioxidant and probiotic foods has already become part of our daily life.

In the past few years, it has become progressively evident that some foods are associated with an increase in markers of inflammation and endothelial dysfunction, suggesting a role of some dietary patterns in the pathogenesis of atherosclerosis [29,30].

Other studies have evaluated the correlation between some dietary habits and the risk of developing chronic inflammatory autoimmune diseases, such as rheumatoid arthritis and systemic lupus erythematosus, emphasizing the possible advantages of a diet with restricted salt and omega-6 polyunsaturated fatty acids, but rich in whole grains, green tea, and supplements, such as vitamin D, flax seeds, turmeric, and others [31,32,33].

Furthermore, some authors have investigated the role of certain dietary regimens as a possible additional therapeutic strategy for chronic inflammatory bowel diseases: an unhealthy diet is closely associated with intestinal dysbiosis, low microbial variability, and a deficiency of beneficial commensal bacteria, which lead to alterations of the intestinal mucosa and stimulation of the innate immune system. These findings underpin the concept of immunonutrition, which is a therapeutic strategy that aims to re-establish intestinal homeostasis with foods or supplements with an anti-inflammatory and immuno-modulatory effect [34,35,36].

FMF patients may have a different gut microbiota composition compared to healthy subjects, and several studies have recently explored the relationship between FMF clinical features, colchicine effectiveness and intestinal microbiota [37,38,39]. Both genetic factors and the persistent autoinflammatory status per se play a crucial role in determining the gut microbiota profile in subjects with FMF and its composition may even influence the development of amyloidosis [40].

Some authors have described cases of more severe and frequent attacks in FMF patients with *Helicobacter pylori* infection, where its eradication led to a reduction in attacks and cytokine levels [41,42].

Verrecchia et al. showed that small intestinal bacterial overgrowth (SIBO) may affect FMF clinical severity, and patients with a reduced or no responsiveness to colchicine may benefit from SIBO investigation, since appropriate antibiotic therapy can improve the effectiveness of colchicine and reduce the use of second-line, more expensive drugs [43].

Since a deficiency or an excess of certain macro- and micronutrients greatly affect the functions of the immune system, a comprehensive dietary program, with an elevated anti-inflammatory/pro-inflammatory ratio, may play an important role in controlling FMF acute attacks. Consequently, the association between diet and FMF is drawing progressively more research attention.

To date, little is known about how diet affects the severity of the clinical phenotype of FMF. However, these patients often report a correlation between clinical relapse and the intake of particular foods which, for this reason, are eliminated from their diet.

The first study to investigate the role of diet in triggering FMF acute attacks was conducted by Mellinkoff et al., who showed that a diet restricted to 20 g-of-fat reduced the incidence of attacks and prompt exacerbations of the disease closely followed dietary indiscretions [44]. However, the true benefit of this diet was controversial and not confirmed in other, subsequent studies [45,46,47].

According to Schwabe and Peters, one of the reasons for the varying benefit of the 20 g fat diet in different studies was poor patient adherence. In their retrospective study, the authors analyzed the recurrence of FMF attacks in a group of 46 patients that had been on Mellinkoff’s 20-g-of-fat diet for two or more years. The putative beneficial effect of the low-fat diet was evaluated by self-reported symptoms and the use of an unvalidated questionnaire. 28 patients discontinued the diet because it was considered too rigid. Thus, contrasting with the original conclusions reported by Mellinkoff et al., Schwabe and Peters found no statistically significant difference in the frequency of attacks between those following the 20-g-of-fat diet and those who were not [21].

Yenokyan and Armenian also found a negative, statistically significant relation between the consumption of high-fat-containing foods and the likelihood of having FMF attacks. The rationale underlying this finding is still controversial, although it seems possibly due to a loss of appetite and low intake of any food prior to acute attacks [23].

Ekinci et al. found a statistically significant higher rate of complete colchicine response in children who preferred less salty or less fatty meals, according to the Children’s Dietary Self-Efficacy Scale (CDSS) and Health Behaviour Questionnaire (HBQ)—Diet Behaviour Scale (DBS). Therefore, lower salt and fat intake may reduce the inflammatory burden and lead to a milder disease course in FMF patients [24].

Since milk is the main food consumed in childhood, one study evaluated whether breastfeeding could have an impact on the severity of the disease, given its important role in the development of innate immunity. The mothers of 81 FMF children were asked to complete a questionnaire, and no correlation was found between the duration of breastfeeding and severity of the clinical picture of FMF in children. Nor was a correlation shown with the age at which cow’s milk and other supplements were introduced [22].

Şentürk et al. showed that in a Turkish cohort of 1119 FMF patients, more than half used at least one type of complementary and integrative therapy (CIT) (i.e., vitamins, mineral, and nutritional supplements) to control their symptoms, with or without FMF-specific therapy. Most of the patients who used CIT were women suffering from joint pain, gastrointestinal symptoms, and dyspnea; 32.8% used mineral supplements and vitamins, such as calcium, iron, and vitamins B_12_, C, D, and E; 25% used nutritional supplements, such as fish oil and honey; and 24.6% used herbs, such as ginger, green tea, turmeric and rosemary. Authors reported a statistically significant relationship between having five or more health check-ups per year and CIT use, suggesting that individuals who are very aware of their own health seem to be more likely to use CIT for symptom management. At the same time, a statistically significant negative relationship was found between hospitalization due to FMF and CIT use. Unfortunately, the putative health-related beneficial effects of CITs were not investigated (specifically, no information was recorded about way of use), and no further conclusions could be drawn from these results [27].

As mentioned above, vitamins C and D, omega-3 polyunsaturated fatty acids contained in fish oil, green tea, and turmeric, together with others, such as vitamin E and zinc, belong to the category of immunonutrients. Thanks to their anti-inflammatory and antioxidant effects, these nutrients counteract the inhibitory effect of inflammation on T lymphocytes. Therefore, by positively affecting the immune system, they could improve the clinical status of patients.

In addition, Kazem et al. investigated the effects of an anti-inflammatory diet (rich in fresh vegetables and fruits, low in saturated and unsaturated fats and carbohydrates, low in food additives, sugar, fast foods, and processed foods) and nutritional supplements (vitamin D, curcumin, and flaxseed) in 73 patients on colchicine therapy. After six months of dietary intervention, a statistically significant improvement in symptoms (evaluated by self-reported well-being), cognitive tests, e.g., the Mini-Mental State Examination, and in C-reactive protein serum levels was observed [26].

Finally, Carroccio et al. demonstrated that wheat ingestion can lead to immune activation and exacerbation of FMF. In a double-blind placebo-controlled trial, based on a challenge with wheat, the authors observed that patients showed a higher Auto-Inflammatory Diseases Activity Index (AIDAI) score than at baseline or with placebo [48]. Moreover, mean serum C-reactive protein and serum amyloid A levels almost doubled in FMF patients after the wheat challenge, but this did not reach statistical significance. Moreover, the number of circulating CD14^+^/IL-1β^+^ and CD14^+^/TNF-α^+^ monocytes increased significantly. This study, however, must be considered a pilot study as it involves a very limited number of patients, and its results need to be confirmed [25]. The authors suggested that a specific non-gluten protein component of wheat, the family of amylase/trypsin-inhibitors (ATIs), might activate innate immunity and inflammatory processes in the gut of FMF patients. ATIs engage the toll-like receptor 4 (TLR4)-MD2-CD14 complex, leading to an upregulation of monocyte-, macrophage-, and especially dendritic cell maturation- markers and an increased release of pro-inflammatory cytokines and chemokines by these myeloid cells [49].

In this context, however, it is not possible to rule out that a “simple” distension effect of the intestinal loops induced by the fermentable, oligo-, di-, monosaccharides and polyols (FODMAPs) contained in wheat (e.g., fructans), on top of a condition of persistent intestinal autoinflammatory status, might be the culprit behind the worsening of the patients’ reported symptoms.

In view of the findings of this narrative review, it is intuitive that further studies are required to establish the actual role of diet in managing FMF symptoms. To support this hypothesis, some experimental studies, mainly conducted in vitro or using murine models, have shown that an excessive consumption of salty foods can stimulate the immune system towards a pro-inflammatory response, with the overexpression of IL-1β, which is currently the target of second-line drugs for FMF, namely canakinumab and anakinra [33,50]. Moreover, other studies have shown that the intake of ancient grains could reduce the production of circulating proinflammatory cytokines [51,52], probably also thanks to their lower concentration of ATI and FODMAPs compared to modern grains [53,54].

## 5. Conclusions

The correlation between diet and the clinical course of FMF is still unclear and needs further evaluation. The use of a sustainable diet, without excessive restrictions, rich in antioxidants and supplements with an anti-inflammatory effect, can be an auxiliary therapeutic strategy that can improve the quality of life of FMF patients. Our study group aims, in the near future, to further investigate the relationship between the ingestion of wheat and the exacerbation of acute FMF attacks and the effects on responsiveness to colchicine.

## Figures and Tables

**Figure 1 nutrients-14-03216-f001:**
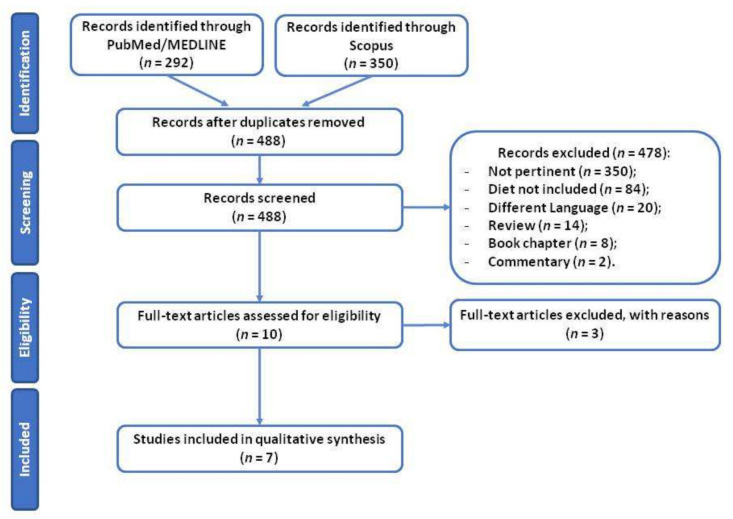
Flowchart representing the PRISMA flow diagram of the screening and selection of the studies.

**Table 1 nutrients-14-03216-t001:** Characteristics and main results of the included studies (reported in order of publication year).

Reference Article [Reference No]	Publication Year	Year(s) of Study	Country	Ethnicity	Study Design	Primary Objective of the Study	No. of Participants	Outcome	Food Considered	Findings and Interpretations
Schwabe, A.D. [21]	1974	1960–1972	USA	Armenian	Retrospective study	To review the clinical manifestations, complications, and prognosis in 100 Armenians with FMF followed for 2–12 years. To assess differences in FMF attack recurrence in a subgroup of 46 patients undergoing the 20-g-of-fat daily diet.	100	FMF attack recurrence	Fatty foods (sausage, pork, eggs, and ice-cream); alcohol	No conclusions can be drawn from this limited study. Adherence to the 20-g-of-fat daily diet represented the major problem in these patients since only 18 out of 46 patients fully followed the above-mentioned diet for 2 years, without statistically significant results.
Makay, B. [22]	2009	2009	Turkey	Turkish	Retrospective study	To investigate whether being breastfed and duration of breastfeeding has an impact on the phenotypic expression of FMF	81	FMF severity	Breastfeeding, formula feeding, cow’s milk feeding, complementary feeding	Breastfeeding is not an exogenous factor having an impact on FMF disease severity. Further collaborative studies on large series from different geographic regions investigating the effect of breastfeeding on severity of FMF are required.
Yenokyan, G. [23]	2012	2007–2008	Armenia	Armenian	Case-crossover study	To estimate if stressful events, like a high-fat diet, represent a trigger for FMF attacks in a restricted exposure window	167	Time between consumption of high-fat foods and FMF attacks	High-fat diet (beef, pork, other lunch meat, butter, mayonnaise, eggs, cheese, milk, popcorn, French fries, sour cream, yogurt, ice cream, pastry)	Statistically significant negative relation between consumption of high-fat-containing food items and the likelihood of developing FMF attacks.
Ekinci, R. [24]	2020	2019	Turkey	Turkish	Retrospective cross-sectional study	To assess diet behaviors and self-efficacy in children with FMF and the relation with symptoms, attack frequency, and treatment outcomes	74	Distribution of MEFV mutations, CDSS and DBS scores pooled and relation with foods	High-fat and high-salt foods	Statistically significant higher rate of complete colchicine response in patients with a preference for less salty or fatty meals. The symptoms and laboratory results did not differ between patients grouped according to their dietary self-efficacy and behaviors.
Carroccio, A. [25]	2020	2015–2017	Italy	Italian	Case-crossover study	(1) To determine if a 2-week double-blind placebo-controlled (DPBC) crossover wheat vs. rice challenge exacerbates clinical manifestations of FMF; (2) To evaluate the innate response of non-celiac wheat sensitivity (NCWS)/FMF patients who underwent the DPBC challenge	6	(1) Clinical symptoms, by an FMF-specific AIDAI (Auto-Inflammatory Diseases Activity Index) score; (2) Serum soluble CD14 (sCD14), C-reactive protein (CRP), and serum amyloid A (SSA); (3) Circulating CD14^+^ monocytes expressing IL-1β and TNF-α	Wheat	The AIDAI score significantly increased in FMF patients during DBPC with wheat, but not with rice (19 ± 6.3 vs. 7 ± 1.6; *p* = 0.028). sCD14 values did not differ in FMF patients before and after the challenge but were higher in FMF patients than in healthy controls (median values 11357 vs. 8710 pg/mL; *p* = 0.002). The percentage of circulating CD14^+^/IL-1β^+^ and CD14^+^/TNF-α^+^ monocytes increased significantly after DBPC with wheat vs. baseline or rice challenge.
Kazem, Y. [26]	2020	2017–2018	Egypt	Egyptian	Retrospective and cross-sectional study (before/after)	To highlight the effect of an anti-inflammatory diet, containing vitamin D, curcumin and flaxseed supplementation, on the clinical presentation, general well-being and cognitive functions of a group of FMF patients	73	FMF attack recurrence, subjective well being	Anti-inflammatory diet (rich in fresh vegetables and fruits, low in saturated and unsaturated fats and carbohydrates, low in food additives, sugar, fast foods and processed foods) + dietary supplementation with vitamin D, curcumin and flaxseeds	The anti-inflammatory diet, containing vitamin D, curcumin and flaxseed supplementation, ameliorated the clinical presentation, general well-being and cognitive functions of FMF patients
Şentürk, S. [27]	2021	2018–2019	Turkey	Turkish	Retrospective and cross-sectional study	To evaluate correlations between the use of complementary and integrative therapies (CIT) and the symptoms of Turkish patients with FMF. The study sought to answer the following questions: (1) What is the frequency of CIT use in disease management in Turkish patients with FMF? (2) What CIT modalities are used by Turkish patients with FMF? (3) Is there a relationship between CIT use and symptoms in Turkish patients with FMF?	1119	FMF-related hospitalization	Mineral supplements and vitamins (calcium, iron, zinc, and vitamin B12, C, D and E); nutritional supplements (fish oil, honey, and ginseng pills)	Statistically significant relationship between having five or more health check-ups per year and CIT use. Individuals who are very aware of their own health seem to be more likely to use CIT for symptom management, suggesting that individuals with FMF effectively benefit from health care. At the same time, a statistically significant negative relationship was found between hospitalization due to FMF and CIT use. This result suggests that individuals with FMF used CIT methods in an uncontrolled manner. However, as the questionnaire did not ask how CIT methods were used, this relationship may be more complicated.

## Data Availability

All data are presented in the current manuscript (text and tables).

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
