# Peer review of "Familial Mediterranean Fever and Diet: A Narrative Review of the Scientific Literature"

_nutrients, 2022, doi:10.3390/nu14153216_

Round 1
Reviewer 1 Report
In this review, Mansueto and colleagues describe the findings of seven papers investigating the effects of nutrition on FMF. Overall, the review is informative and well-written, but I have a couple of suggestions for improvement of the manuscript. Most importantly, I believe the authors should be a bit more careful in the abstract and the discussion when drawing generalized conclusions about the seven papers they discuss.
Major remarks:
1. There are a number of inaccuracies in the introduction of this review:
- Line 44-45: ‘pyrin is commonly expressed in neutrophils’. This is true, but the sentence raises the impression that pyrin is only/mainly expressed in neutrophils, while its expression pattern is broader than only neutrophils and thus may play a role in FMF pathogenesis also by acting in other cell types. Please describe the expression of pyrin more accurately.
- Line 45-48: ‘its mutations lead to an excessive production and secretion of interleukin 1 (IL-1) and other inflammatory cytokines, activation of the NF-кB pathway, caspase-1 and of the apoptosis process’. Please specify ‘IL-1’ to IL-1beta. ‘Other inflammatory cytokines’ raises the impression that pyrin induces several cytokines, while in fact it only directly triggers IL-1beta and IL-18 release and other cytokines are indirectly upregulated due to the inflammatory response. Please specify. Pyrin does not activate NF-kB directly, this may occur in FMF patients but it is caused by the increased cytokine levels, not due to pyrin mutations. Please correct. Also the ‘apoptosis process’ is not something that is induced by pyrin. The type of cell death that is induced by pyrin is termed pyroptosis. Please correct.
- Line 62: ‘some foods have been suggested as potential triggers of FMF attacks’ needs references and specifics about which types of food have been suggested.
2. The conclusion about ref 44 on lines 188-191 contradicts with the statement about this paper in the table. The text says there is an effect of the diet, the table says that the results were not statistically significant thus no effect. Please correct.
3. Lines 223-224 state that ‘no conclusions could be drawn’ about ref 28, while the table mentions several statistically significant results of that paper. This is confusing. Please specify. It would also be helpful for the non-specialist reader to explain better what ‘complementary and integrative therapy’ is exactly.
4. Most importantly, the conclusions that authors themselves draw in lines 26-31 of the abstract are conflicting. They first state that low-fat may be beneficial (referring to ref 44 that according to their table did not have statistically significant results), then state that there are conflicting results and that therefore further study is required, and then nevertheless state that low fat could represent an additional therapeutic strategy for FMF. This does not give a comprehensive message for the reader. Please do an effort to draw a more balanced and correct conclusion from your narrative review.
Minor remark:
1. Typo in the table in discussing ref 25: MEVF should be MEFV.
Author Response
We thank the reviewer for the critical point of view and for the hints he gave us to improve our review. We tried to answer point by point to all the queries.
Q1. There are a number of inaccuracies in the introduction of this review:
We modified the introduction according to the reviewer reports.
Q1.1 - Line 44-45: ‘pyrin is commonly expressed in neutrophils’. This is true, but the sentence raises the impression that pyrin is only/mainly expressed in neutrophils, while its expression pattern is broader than only neutrophils and thus may play a role in FMF pathogenesis also by acting in other cell types. Please describe the expression of pyrin more accurately.
Q1.2 - Line 45-48: ‘its mutations lead to an excessive production and secretion of interleukin 1 (IL-1) and other inflammatory cytokines, activation of the NF-кB pathway, caspase-1 and of the apoptosis process’. Please specify ‘IL-1’ to IL-1beta. ‘Other inflammatory cytokines’ raises the impression that pyrin induces several cytokines, while in fact it only directly triggers IL-1beta and IL-18 release and other cytokines are indirectly upregulated due to the inflammatory response. Please specify. Pyrin does not activate NF-kB directly, this may occur in FMF patients but it is caused by the increased cytokine levels, not due to pyrin mutations. Please correct. Also the ‘apoptosis process’ is not something that is induced by pyrin. The type of cell death that is induced by pyrin is termed pyroptosis. Please correct.
Response
Thank you for this accurate analysis; obviously pyrin has a broader spectrum of expression than neutrophils and we understand our sentence could create a misunderstanding especially in readers who approach for the first time this condition. Moreover, we fully understand the mistake we do trying to synthetize the pathogenetic model of FMF. Thus, we modified the sentences as follow:
“Pyrin is commonly expressed by neutrophils, as well as other cells belonging to the innate immune system: eosinophils, monocytes, and dendritic cells; the pyrin mutation might contribute to the activation of the pyrin inflammasome. Pyrin usually interacts with an adaptor protein, the apoptosis-associated speck-like protein (ASC), and this complex operates as a caspase-1 recruiter, creating the inflammasome, which activates pro-interleukin(IL)-1beta and pro-IL-18 in IL-1beta and IL-18. Furthermore, the pyrin inflammasome can activate the caspase-1 mediated cleavage of gasdermin D (GSDMD) pyroptosis (a peculiar type of cellular death characterized by cell swelling and lysis), with a further release of IL-1beta and IL-18. These overexpression and release of IL-1beta and IL-18 works as an initiator and amplifier of innate immune response, which flows in an uncontrolled proinflammatory cytokines cascade, consequential activation of NF-кB pathway, and auto-amplification of inflammatory processes [2,6–11]”.
Q1.3 - Line 62: ‘some foods have been suggested as potential triggers of FMF attacks’ needs references and specifics about which types of food have been suggested.
Response
Thank you to make us note the lack of reference. We modified the sentence as follow:
“Additionally, some foods have been suggested as potential triggers of FMF attacks (e.g., fatty and salty foods, cow’s milk, formulas, wheat) [21-25] and, for this reason, researchers are looking for a link between certain diet regimens and the severity of the clinical manifestations of FMF or patient response to colchicine therapy [21-27]”.
We modified reference’s number throughout the text.
Q2. The conclusion about ref 44 on lines 188-191 contradicts with the statement about this paper in the table. The text says there is an effect of the diet, the table says that the results were not statistically significant thus no effect. Please correct.
Response
Unfortunately, there has been a misunderstanding in this point. The study to which referees the citation 44 was performed by Mellinkoff et al. in 1961 and it has not been included in our review, as reported in Materials and Methods (“The inclusion criteria for the selected studies were: publication date between 1971 and 2022, English language, and analysis of the association between FMF and diet”). Thus, the study is not reported in the table. We reported this study in the text to introduce the one by Schwabe and Peters (both in the text and in the table) (new reference number 21), which retrospectively analyzed a subgroup of Mellinkoff’s patients, reporting contrasting results. To clarify any misunderstanding, we modified the text as follow:
“Thus, contrasting with the original conclusions reported by Mellinkoff et al., Schwabe and Peters found no statistically significant difference in the frequency of attacks between those following the 20-g-of-fat diet and those who were not [21]”.
Q3. Lines 223-224 state that ‘no conclusions could be drawn’ about ref 28, while the table mentions several statistically significant results of that paper. This is confusing. Please specify. It would also be helpful for the non-specialist reader to explain better what ‘complementary and integrative therapy’ is exactly.
Response
Thank-you so much for helping us to improve clarity of the text. We added both a little sentence about what complementary and integrative therapy includes, and some sentences explain better the results (significant result and impossibility to draw definitive conclusion).
“Åžentürk et al. showed that in a Turkish cohort of 1,119 FMF patients, more than half used at least one type of complementary and integrative therapy (CIT) (i.e., vitamins, mineral, and nutritional supplements) to control their symptoms, with or without FMF-specific therapy. Most of the patients who used CIT were women suffering from joint pain, gastrointestinal symptoms, and dyspnea; 32.8% used mineral supplements and vitamins, such as calcium, iron, and vitamins B12, C, D and E; 25% used nutritional supplements, such as fish oil and honey; and 24.6% used herbs, such as ginger, green tea, turmeric, and rosemary. Authors reported a statistically significant relationship between having five or more health check-ups per year and CIT use, suggesting that individuals who are very aware of their own health seem to be more likely to use CIT for symptom management. At the same time, a statistically significant negative relationship was found between hospitalization due to FMF and CIT use. Unfortunately, the putative health-related beneficial effects of CITs were not investigated (specifically, no information was recorded about way of use), and no further conclusions could be drawn from these results [27]”.
Q4. Most importantly, the conclusions that authors themselves draw in lines 26-31 of the abstract are conflicting. They first state that low-fat may be beneficial (referring to ref 44 that according to their table did not have statistically significant results), then state that there are conflicting results and that therefore further study is required, and then nevertheless state that low fat could represent an additional therapeutic strategy for FMF. This does not give a comprehensive message for the reader. Please do an effort to draw a more balanced and correct conclusion from your narrative review.
Response
We are glad you focused attention on this point. Effectively, the conclusions we reported in abstract could be misleading. We therefore welcomed this observation with great gratitude and modified the text to provide a more balanced conclusion:
“Conclusions: to date conflicting results have been reported about fatty and salty food intake and FMF attackrecurrence; moreover, some authors suggested a possible role of wheat. Finally, a diet rich in antioxidants and supplements with an anti-inflammatory effect could, partially, reduces symptoms and improves well-being of FMF patients. Nevertheless, no conclusive data could be drawn about the impact of diet in FMF symptom’s triggering, and further studies are required to clarify this putative association”.
Minor remark:
- Typo in the table in discussing ref 25: MEVF should be MEFV.
Response
Thank you for allowing use to correct this mistake. We modified the table with the correct abbreviation.

Reviewer 2 Report
In the present study Mansueto et al, investigated the correlation of diet and familial Mediterranean fever from the published reports from 1974 and 2022. The overall design of the study, results interpretation and discussion part has been written nicely. Although a firm and conclusive outcome could not be obtained from the present review but this manuscript can help to devise a new study to explore more about the modulation of FMF though dietary changes.
Author Response
We would like to thank the reviewer for the positive comments about our paper; we are happy that he/she completely understood our aims and the effort we made to understand the impact that diet can determine in a pathological condition as complex as familial Mediterranean fever.

Round 2
Reviewer 1 Report
Thank you for addressing my questions. I accept the paper since the authors have modified the manuscript, albeit in a sloppy manner. E.g. the spelling and grammatical errors in the new abstract:
line 29: reduces, improves?
line 29-30: to draw data?
line 30: symptom's?
Congratulations to the authors.